# Learning Cross-Domain Correspondence for Control with Dynamics Cycle-Consistency

**Qiang Zhang**
Shanghai Jiao Tong University
zhangqiang2016@sjtu.edu.cn

**Tete Xiao**
UC Berkeley
txiao@eecs.berkeley.edu

**Alexei A. Efros**
UC Berkeley
efros@eecs.berkeley.edu

**Lerrel Pinto**
New York University
lerrel@cs.nyu.edu

**Xiaolong Wang**
UC San Diego
xiw012@ucsd.edu

## Abstract

At the heart of many robotics problems is the challenge of learning correspondences across domains. For instance, imitation learning requires obtaining correspondence between humans and robots; sim-to-real requires correspondence between physics simulators and the real world; transfer learning requires correspondences between different robotics environments. This paper aims to learn correspondence across domains differing in representation (vision vs. internal state), physics parameters (mass and friction), and morphology (number of limbs). Importantly, correspondences are learned using unpaired and randomly collected data from the two domains. We propose *dynamics cycles* that align dynamic robot behavior across two domains using a cycle-consistency constraint. Once this correspondence is found, we can directly transfer the policy trained on one domain to the other, without needing any additional fine-tuning on the second domain. We perform experiments across a variety of problem domains, both in simulation and on real robot. Our framework is able to align uncalibrated monocular video of a real robot arm to dynamic state-action trajectories of a simulated arm without paired data. Video demonstrations of our results are available at: https://sjtuzq.github.io/cycle_dynamics.html.

## 1 Introduction

Humans have a remarkable ability to learn motor skills by mimicking behaviors of agents that look and act very differently from them. For example, developmental psychologists has shown that 18-month-old children are able to infer the intentions and imitate behaviors of adults (Meltzoff, 1995). Imitation is not easy: children likely need to infer correspondences between their observations and their internal representations, which effectively aligns the two domains. Learning such a cross-domain correspondence is particularly valuable for robotics and control. For example, in imitation learning, if we want robots to imitate the motor skills of humans (or robots with different morphologies), we need to find the correspondence in both visual observations and morphology dynamics. Similarly, when transferring a policy trained in simulation to a real robot, we, again, need to align visual inputs and physics parameters across different environments.

To align the skills across different domains, several prior approaches have proposed learning invariant feature representations across the domains (Gupta et al., 2017; Sermanet et al., 2018). Policies or visual representations are trained to be invariant to the changes which are irrelevant to the downstream task, while maintaining useful information for cross-domain alignment. However, these methods require paired and aligned trajectories, usually collected by pre-trained policies or human labeling, which is often too expensive to collect for real-world learning problems. Additionally, invariance is a rather strong constraint, and might not be universally suitable. The reason is that different invariances might be beneficial for different downstream tasks, which has been recently studied in self-supervised visual representation learning (Tian et al., 2020).

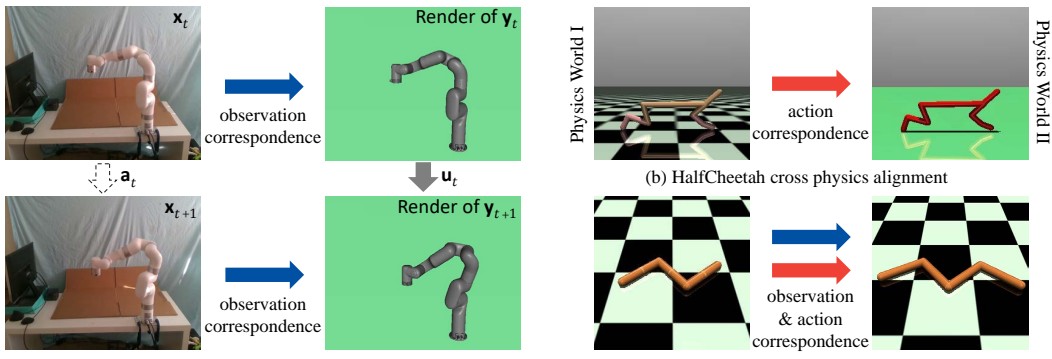

Figure 1: We propose to learn observation correspondence (blue arrow) and action correspondence (red arrow) across domains using Dynamics Cycle-Consistency. Our applications include: (a) Aligning real robot images with simulation states; (b) Aligning actions between environments with different physics parameters (We use different rendering to indicate that the physics are different); (c) Aligning both actions and observations between agents at the same time with different morphology.

Instead of learning invariances, an emerging line of research focuses on finding correspondences by learning to translate between two different domains with *unpaired* data (Zhu et al., 2017; Bansal et al., 2018). While this translation technique has shown encouraging results in imitation learning (Smith et al., 2019) and sim-to-real transfer (Hoffman et al., 2017; James et al., 2019), it is limited to finding correspondences only in the visual observation space. However, in real-world applications, besides visual observations, the physics parameters and morphology dynamics between two domains are also often misaligned. Hence, solely learning with passive visual correspondences, one is unable to reason about the effects of dynamics. We must go beyond the image space and explicitly incorporate dynamics information to truly extend correspondence learning to aligning behaviors.

In this paper, we take the first steps toward learning correspondences which can align behaviors on a range of domains including different modalities (vision vs. agent state), different physical parameters (friction and mass), and different morphologies. Importantly, we use *unpaired* and *unaligned* data from the two domains to learn the correspondences. Specifically, we propose to find *observation correspondences* and *action correspondences* at the same time using dynamics cycle-consistency. Our dynamics cycles chain the observations and actions across time and domains together. The consistency in the dynamics cycle indicates consistent translation and prediction results. The input data to our learning algorithm takes the form of 3-element tuples from both domains: current state, action and the next state. Figure 1(a) exemplifies our model, which is a 4-cycle chain containing the observations of one domain $(\mathbf{x}_t, \mathbf{x}_{t+1})$ (real robot in Figure 1(a)) at two time steps, and another domain $(\mathbf{y}_t, \mathbf{y}_{t+1})$ (simulation in Figure 1(a)). To form a cycle, we learn a domain translator $G : \mathbf{x}_t \mapsto \mathbf{y}_t$ to translate images to states and a predictive forward dynamics model in state space $F : \mathbf{y}_t \times \mathbf{u}_t \mapsto \mathbf{y}_{t+1}$ where $\mathbf{u}_t$ represents the action taken at time $t$, and $\mathbf{a}_t$ is the corresponding action in the real robot domain. The forward model in the real robot domain is not necessary in our framework. The training signal is: given observations in time $t$, the future prediction in time $t + 1$ should be consistent under the consistent action taken across two domains, namely dynamics cycle-consistency.

We explore applications both in simulation and with a real robot. In simulation, we adopt multiple tasks in the MuJoCo (Todorov et al., 2012) physics engine, and show that our model can find correspondence and align two domains across different modalities, physical parameters (Figure 1(b)), and morphologies (Figure 1(c)). Given the alignment, we can transfer a reinforcement learning (RL) policy trained in one domain directly to another domain without further optimizing the RL objective. For our real robot experiments, we use the xArm Robot (Figure 1(a)). Given only uncalibrated monocular videos of the xArm performing *random* actions, our method learns correspondences between the real robot and simulated robot without any paired data. At test time, given a video of the robot arm executing a smooth trajectory, we can generate the same trajectory in simulation.

## 2 RELATED WORK

**Learning invariant representations.** To find cross-domain alignment, researchers have proposed to learn representations which are invariant to the changes unrelated to the downstream task (Tobin et al., 2017; Peng et al., 2018; Gupta et al., 2017; Sermanet et al., 2018; Liu et al., 2017b; Pinto et al.,

2017; Sadeghi & Levine, 2016; Yan et al., 2020; Chen et al., 2020; Andrychowicz et al., 2018). For example, domain randomization (Tobin et al., 2017; Sadeghi & Levine, 2016; Andrychowicz et al., 2018; Ramos et al., 2019; Zakharov et al., 2019; Wu et al., 2019) aligns the simulated and real world for policy transfer. However, it assumes that differences between two domains can be covered by hand-crafted augmentations, which may not hold when the domain changes happen to lie outside of these assumptions. To align two domains where the dynamics are different, Gupta et al. (2017) propose to learn invariant features with pairs of states from two domains. However, paired data is hard to collect, and the method is limited to state space, while real-world observations are often based on images (Taylor & Stone, 2009).

**Learning translation.** Instead of learning invariance, our method is related to works which learn the mapping across two domains for alignment (Taylor et al., 2007; Ammar et al., 2015; Tzeng et al., 2015; Joshi & Chowdhary, 2018; Kim et al., 2019; Smith et al., 2019). For example, Tzeng et al. (2015) design an approach to weakly align pairs of images in the source and target domains. Given the paired images, they can perform adaption from simulation to real robot. However, this work has only focused on translation between visual observations. Going beyond visual adaptation, Ammar et al. (2015) utilize unsupervised manifold alignment to find correspondence between states across domains from demonstrations. However, this method uses hand designed features, which restricts its generalization ability. Kim et al. (2019) propose imitation learning with unpaired and unaligned demonstrations. While with less constraint, it requires a trained RL policy to collect demonstrations in both domains for training, and RL is involved in the correspondence learning process. This leads to learning correspondence only relevant to a specific task. In contrast, most of our experiments assume that *we do not know the downstream task and we do not have access to the rewards for RL*. Hence, it is a more general problem setting and can be used for a variety of applications. Our method can learn correspondence between simulated and real robot through *unpaired and randomly collected* trajectories.

In transfer learning, several works have looked at architectural novelties to improve transfer across RL problems (Parisotto et al., 2015; Pinto et al., 2016; Rusu et al., 2016a; Barreto et al., 2017; Omidshafiei et al., 2017; Rusu et al., 2016b). Our method of using cycle consistency pursues an orthogonal direction of architecture design and is compatible with these approaches.

**Cycle-Consistency.** Our work is inspired by literature on cycle-consistency (Zhou et al., 2016; Zhu et al., 2017; Liu et al., 2017a; Hoffman et al., 2017; Bansal et al., 2018; Bousmalis et al., 2018; James et al., 2019). For example, CycleGAN (Zhu et al., 2017) uses cycle-consistency loss with the Generative Adversarial Networks (Goodfellow et al., 2014) for unpaired image-to-image translation, which is subsequently extended for videos (Bansal et al., 2018) and domain adaptation (Hoffman et al., 2017). Similar techniques are applied in sim-to-real transfer by training a image translation model between simulation and real-world images (Stein & Roy, 2018) or aligning both the simulation and real images to the same canonical space (James et al., 2019). Recently, Rao et al. (2020) propose RL-CycleGAN to perform sim2real image translations by adding an extra supervision signal from the Q function. However, all these works are restricted to visual alignments, while ours can align agents cross different dynamics and structures.

## 3 LEARN CORRESPONDENCE USING DYNAMICS CYCLE-CONSISTENCY

**Problem setup.** We aim to learn correspondence across various domains, i.e., input modalities, physics parameters, and morphology. We formulate the trajectories of domain $X$ and $Y$ as $\tau_X \doteq (\mathbf{x}_t, \mathbf{a}_t, \mathbf{x}_{t+1})$ and $\tau_Y \doteq (\mathbf{y}_t, \mathbf{u}_t, \mathbf{y}_{t+1})$, where $\mathbf{x} \in \mathcal{R}^{n_1}$ and $\mathbf{y} \in \mathcal{R}^{n_2}$ are observation representations in domain $X$ and $Y$, $\mathbf{a} \in \mathcal{R}^{m_1}$ and $\mathbf{u} \in \mathcal{R}^{m_2}$ are action representations in domain $X$ and $Y$, and $t$ is time step. Without loss of generality, we assume to learn correspondence from domain $X$ to domain $Y$. [1] Suppose that we have observation alignment functions $G : X \mapsto Y$, and action alignment function $H : X \times A \mapsto U$ and its inverse counterpart $H^{-1}$ as a function $P : Y \times U \mapsto A$. We define two types of correspondence as follows.

*Observation Correspondence*, i.e., what the representation of one observation in domain $X$ should correspond to if it is in domain $Y$, and vice versa. For example, if $X$ is visual sensing of an agent while $Y$ is the state (e.g., joint angle) of the *same* agent, $G$ functions as a state estimator. If $X$ is the

---

[1]We will subsequently show that for various applications only action mapping ought to be bidirectional, whereas observation mapping can be unidirectional.

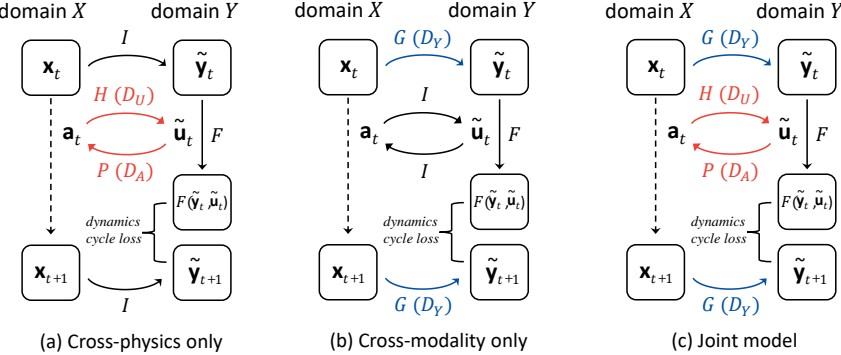

Figure 2: **Model framework**: (a) Model for only *cross-physics* alignment; (b) Model for only *cross-modality* alignment; (c) Joint model for *cross-modality-and-physics* alignment. Red arrows indicate correspondences between actions and blue arrows indicate correspondence between observations.

state of one agent while $Y$ is the state of a *structurally different* agent, such as a Sawyer arm and a UR5 arm, $G$ aligns the states at a same stage towards a common goal (e.g., robot joint positions). We denote two correspondent observations between $X$ and $Y$ as $\mathbf{x} \Leftrightarrow \mathbf{y}$.

*Action Correspondence*, i.e., with correspondent initial observations which actions to execute so that the next observations in two domains remain correspondent. For example, if $X$ and $Y$ are two environments with different physics parameters, with the initial observations $\mathbf{x}_t$, $\mathbf{y}_t$ and $\mathbf{x_t} \Leftrightarrow \mathbf{y_t}$, after action $\mathbf{a}_t$ is executed in domain $X$ and leads to the next observation $\mathbf{x}_{t+1}$, alignment function $H$ should find the action $\mathbf{u}_t$ which leads to next observation $\mathbf{y}_{t+1}$ in domain $Y$ where $\mathbf{x}_{t+1} \Leftrightarrow \mathbf{y}_{t+1}$, and vice versa for $H^{-1}$. We denote two correspondent actions from $X$ and $Y$ as $(\mathbf{x}_t, \mathbf{a}_t) \Leftrightarrow (\mathbf{y}_t, \mathbf{u}_t)$.

Learning observation correspondence and action correspondence enables estimating states from visual input, adapting to environments with different physics, and being able to function even when the structure of the agent changes.

**Method.** We begin by simply mapping states across domains by adversarial training. Given unpaired samples $\{\mathbf{x}_i\} \in X$, and $\{\mathbf{y}_i\} \in Y$, a mapping function $G$ can be learned with a discriminator $D_Y$ with the adversarial objective, where $G$ tries to map $\mathbf{x}$ onto the distribution of $\mathbf{y}$, while $D_Y$ tries to distinguish translated samples $G(\mathbf{x})$ against real samples $\mathbf{y}$:

$$\min_G \max_{D_Y} \mathcal{L}_{\text{adv}}(G, D_Y) = \mathbb{E}_{\mathbf{y}\sim p(\mathbf{y})}\left[\log D_Y(\mathbf{y})\right] + \mathbb{E}_{\mathbf{x}\sim p(\mathbf{x})}\left[\log\left(1 - D_Y(G(\mathbf{x}))\right)\right] \quad (1)$$

The adversarial objective reaches global optimal when the mapping function $G$ can perfectly ground the translated samples onto the distribution defined by $\{y_i\}$.

We learn an action mapping function $H : X \times A \mapsto U$ which maps actions from domain $X$ to domain $Y$, and model its inverse counterpart $H^{-1}$ as a function $P : Y \times U \mapsto A$ with separate parameters. Besides using two adversarial losses with discriminators $D_U$ in $Y$ and $D_A$ in $X$, i.e., $\mathcal{L}_{\text{adv}}(H, D_U)$ and $\mathcal{L}_{\text{adv}}(P, D_A)$, we add cross-domain cycle consistency loss (Zhu et al., 2017) into the objective:

$$\min_{H,P} \mathcal{L}_{\text{dom\_cyc}}(H, P) = \mathbb{E}_{\mathbf{a}\sim p(\mathbf{a})}\left[\left\|P(\mathbf{y}, H(\mathbf{x}, \mathbf{a})) - \mathbf{a}\right\|_1\right], \quad (2)$$

which implies that the translated action should be able to be translated back: $P(\mathbf{y}, H(\mathbf{x}, \mathbf{a})) \approx \mathbf{a}$.

Nevertheless, the structure of learnt mapping by adversarial training is loosely constrained. Vanilla adversarial training may map all samples $X$ to a few samples of $Y$, which still minimizes the adversarial objective. Adding domain cycle consistency loss does not solve the problem fundamentally: for example, given two correspondent but *unpaired* observations, i.e., $\mathbf{x}_t, \mathbf{y}_t$ and $\mathbf{x}_{t+1}, \mathbf{y}_{t+1}$, $G$ can map $\mathbf{x}_t$ to $\mathbf{y}_{t+1}$ and $G^{-1}$ can still map $\mathbf{y}_{t+1}$ back to $\mathbf{x}_t$, which does not violate domain cycle-consistency.

Beyond only relying domain cycle consistency, we exploit the transition dynamics of two domains, termed as *dynamics cycle-consistency*. As illustrated in Figure 2(c), we map the observation-action pair at time step $t$ $\mathbf{x}_t$ and $\mathbf{a}_t$ from domain $X$ to $Y$ using $G$ and $H$, then execute the translated observation and action $\tilde{\mathbf{y}}_t$ and $\tilde{\mathbf{u}}_t$) in domain $Y$ by its transition dynamics $T_Y : Y \times U \mapsto Y$ to get the next observation, which is expected to be correspondent to the next observation from domain $X$, i.e., $T_Y(\tilde{\mathbf{y}}_t, \tilde{\mathbf{u}}_t) \Leftrightarrow \mathbf{x}_{t+1}$. According to the definition of observation correspondence, $T_Y(\tilde{\mathbf{y}}_t, \tilde{\mathbf{u}}_t)$ should

be the same as $G(\mathbf{x}_{t+1})$, as expressed in the objective:

$$\min_{G,H} \mathcal{L}(G,H)_{\text{dyn\_cyc}} = \mathbb{E}_{(\mathbf{x}_t, \mathbf{a}_t, \mathbf{x}_{t+1}) \sim p(\tau_X)} \left[ \left\| G(\mathbf{x}_{t+1}) - T_Y(G(\mathbf{x}_t), H(\mathbf{x}_t, \mathbf{a}_t)) \right\|_1 \right]. \quad (3)$$

One obstacle remains. The transition dynamics $T_Y$ in Equation 3 is in fact the physical property of a simulator or the real world, hence it is not differentiable for back-propagation. In consequence, we train a forward model which takes an observation-action pair as input and predicts the next observation to approximate the dynamics of the environment. Since we have access to trajectories from $Y$, we can directly train the forward model using supervised regression objective:

$$\min_F \mathcal{L}_{\text{forward}}(F) = \mathbb{E}_{(\mathbf{y}_t, \mathbf{u}_t, \mathbf{y}_{t+1}) \sim p(\tau_Y)} \left[ \left\| \mathbf{y}_{t+1} - F(\mathbf{y}_t, \mathbf{u}_t) \right\|_1 \right] \quad (4)$$

Note that forward model $F$ is first pre-trained and it is *not* optimized together with the dynamics cycle-consistency objective, as otherwise $G$ and $F$ can learn to map everything to zero so that $L_{\text{dyn\_cyc}}$ becomes zero, which leads to a trivial solution. Consequently, our full objective is:

$$\mathcal{L}_{\text{full}} = \lambda_0 \mathcal{L}_{\text{dyn\_cyc}}(G,H) + \lambda_1 \left( \mathcal{L}_{\text{adv}}(H, D_U) + \mathcal{L}_{\text{adv}}(P, D_A) + \mathcal{L}_{\text{dom\_cyc}}(H, P) \right) + \lambda_2 \mathcal{L}_{\text{adv}}(G, D_Y) \quad (5)$$

where $\lambda_0$, $\lambda_1$ and $\lambda_2$ are constants balancing the losses.

**Optimization.** We collect unpaired trajectories $\tau_X$ and $\tau_Y$ by executing random actions from both domains. Directly optimizing the full objective end-to-end leads to model collapse, as it involves joint optimization with multiple neural networks: $G$ and $H$ can easily discover a "shortcut" solution, where the translated observations and actions are not valid but they can fool the forward model to optimize the dynamics cycle-consistency objective. Since the forward model is only optimized on trajectory data $\tau_Y$, thus we first pre-train the forward model and fix its parameters throughout the following training procedure. We initialize the action mapping function using an algorithm detailed in the Appendix A.4. We pro-

---

**Algorithm 1:** Alternatingly Joint Training Algorithm

**Input:** Domain X: $\tau_X = \{(\mathbf{x}_t, \mathbf{a}_t, \mathbf{x}_{t+1})\}$
Domain Y: $\tau_Y = \{(\mathbf{y}_t, \mathbf{u}_t, \mathbf{y}_{t+1})\}$
// Training Forward Model Stage
train $\mathcal{L}_{\text{forward}}(F)$ (Eq. 4) to learn transition dynamics $T_Y$ in domain Y;
// Alternatingly Training Stage
**for** $i = 1$ *to* $e$ **do**
  reset $\lambda_1$, set $\lambda_2 = 0$; fix weight of $G$;
  **for** $j = 1$ *to* $e_1$ **do**
   | using $\mathcal{L}_{\text{full}}$ (Eq. 5) to train model $H$ and $P$;
  reset $\lambda_2$, set $\lambda_1 = 0$; fix weight of $H$ and $P$;
  **for** $j = 1$ *to* $e_2$ **do**
   | using $\mathcal{L}_{\text{full}}$ (Eq. 5) to train model $G$;
**return** State alignment model $G$
Action alignment model $H$ and $P$

---

pose to employ alternating training procedure for the full objective: When we train the observation mapping function $G$ and its auxiliary discriminator $D_Y$, we fix the action mapping function $H$ and $P$; then when the action mapping function $H$ and $P$ with $D_U$ and $D_A$ are trained, we fix the observation mapping function $G$. Since the action mapping functions are reasonably initialized, at the beginning of training procedure the observation mapping function is optimized. It is grounded on good action mappings, as well as the dynamics of environments by dynamics cycle consistency, thus it is constrained from learning an arbitrary short cut. Subsequently, action mapping functions can be further fine-tuned once we obtain a good observation mapping function (Algorithm 1).

**Tasks.** Our formation of correspondence learning is broad and general, and it enables many applications which typically require intricately designed frameworks or are hard to solve without paired data. Specifically, we study the following three tasks:

The first task is *cross-physics* alignment, where domain $X$ and domain $Y$ are two environments with different *physics parameters* but same input modality. As shown in Figure 2(a), same input modality indicates that observation correspondence always holds, i.e., $\mathbf{x}_t \equiv \mathbf{y}_t$; different physics parameter indicates that executing a same action at the same initial observation in separate environments results in different next observation. After learning correspondences, assuming we have a policy in domain $Y$, we can transfer it to domain $X$ by mapping the predicted action of the policy $\mathbf{u}$ from domain $Y$ to $X$ with action mapping function $P$. The translated action $\tilde{\mathbf{a}}$ can then be executed in domain $X$.

The second task is *cross-modality* alignment, where domain $X$ and domain $Y$ are different sensing (observation) modality of the *same* agent, which implies that action correspondence between two domains always hold (see Figure 2(b)). In other words, $H$ and $P$ are both identity mapping, and $\mathbf{a}_t \equiv \mathbf{u}_t$. Thus we can set $\gamma = 0$ in Eq. 5 in training. A predominant choice is $X$ being image while

$Y$ being state, where $G$ essentially learns to perform state estimation. Moreover, we can execute a policy which is originally trained on *state space* in *image space*, as the input $\mathbf{x}_t$ in image space can be translated by $G$ before fed into the policy based on state space, yielding a predicted action $\mathbf{u}_t$, which can be directly executed in domain $X$.

Combining the above-discusses two tasks yields the third task, in which cross-physics and cross-modality alignment are realized simultaneously, thanks to our proposed joint alternative training procedure. We refer to it as *cross-modality-and-physics* alignment, as shown in Figure 2(c). This formulation can be further extended to another task, where domain $X$ and $Y$ are two agents with different morphologies, termed as *cross-morphology* alignment. For example, domain $X$ can be a three-leg cheetah and domain $Y$ can be a two-leg cheetah. In this case, the representations of $\mathbf{x}$ / $\mathbf{y}$ and $\mathbf{a}$ / $\mathbf{u}$ are fundamentally different, yet intrinsically they share similarities in locomotion.

As the correspondence is established between two domains, it can be applied to different downstream applications. Suppose that our goal is to transfer a policy trained in domain $Y$ to $X$. Inference includes three steps: (i) Given an observation $\mathbf{x}_t$ in domain $X$, use observation mapping function $G$ to translate $\mathbf{x}_t$ to $\mathbf{y}_t$; (ii) Execute the policy in domain $Y$ given $\mathbf{y}_t$, and obtain the action output $\mathbf{u}_t$; (iii) Translate the action $\mathbf{u}_t$ from domain $Y$ back to domain $X$ with the action mapping function $P$.

**Implementation Details.** The networks $D, F, H, P$ are implemented by MLPs, and network $G$ is a ResNet-18 (He et al., 2016) with a 4-layer MLP head. For the inputs of $G$, instead of using one static image, we concatenate the current frame and two consecutive past frames together to capture any motion information. We first train the forward dynamics model $F$ for 20 epochs using Adam (Kingma & Ba, 2014) with 0.0001 learning rate. We then train the other networks for 50 epochs with the same learning rate. We set $e_1$ and $e_2$ to 5000 steps in Algorithm 1. See Appendix B for more details.

## 4 SIMULATION EXPERIMENTS

We first test the efficiency of our framework and conduct ablation studies in simulation environments. We choose MuJoCo physics simulator as our test bed. We model domain $X$ and $Y$ as two different environments, where input modality, physics parameters, and morphology structures of the agents can vary. We believe that our method can be applied to a lot of environments. However, in this paper we focus on the representative ones including four tasks based on OpenAI Gym (Brockman et al., 2016), i.e., "HalfCheetah", "FetchReach", "Walker" and "Hopper", and one task based on DeepMind Control (Tassa et al., 2018), i.e., "FingerSpin". We perform experiments with different settings including: (i) Cross-physics alignment, where only the physical parameters are different in two domains; (ii) Cross-modality alignment, where only the observation space is different; (iii) Cross-modality-and-physics alignment, a joint task of (i) and (ii); (iv) Cross-morphology alignment, where agent structures in two domains are different. To sample the training data, we randomly collect $50k$ unpaired trajectories in both domain $X$ and domain $Y$ in most settings. The evaluation dataset size is $10k$. Besides evaluating on the alignment errors, we also benchmark how well the pre-trained RL policies in one domain can be transferred to another domain. To pre-train the policy, we use DDPG (Lillicrap et al., 2015) with HER (Andrychowicz et al., 2017) for "FetchReach" and TD3 algorithm (Fujimoto et al., 2018) for other environments. Note that we do *not* need to further fine-tune the policy for transferring to a new domain. We report the task success rate for "FetchReach" and task rewards for the other environments. All RL policies are trained with 5 different seeds. More details about our method implementation and the reference policies can be found in the Appendix B.

**Cross-physics alignment.** In order to create environments with different physics parameters, we modify armature and mass in the environments. We use default armature and torso mass parameters in domain $Y$. To create domain $X$, we increase the armature for tasks including

| Tasks | Oracle, $Y$ | Direct, $Y{\rightarrow}X$ | DR, $Y{\rightarrow}X$ | Ours, $Y{\rightarrow}X$ | Oracle, $X$ |
|---|---|---|---|---|---|
| HalfCheetah | $6270{\pm}123$ | $3651{\pm}665$ | $3763{\pm}752$ | $\mathbf{3997{\pm}438}$ | $6769{\pm}185$ |
| FingerSpin | $804{\pm}89$ | $483{\pm}186$ | $492{\pm}284$ | $\mathbf{562{\pm}124}$ | $765{\pm}68$ |
| FetchReach$^\dagger$ | $100\%$ | $100\%$ | $100\%$ | $\mathbf{100\%}$ | $100\%$ |
| Walker2d | $875{\pm}24$ | $516{\pm}395$ | $546{\pm}258$ | $\mathbf{667{\pm}174}$ | $816{\pm}17$ |
| Hopper | $2364{\pm}635$ | $1542{\pm}1041$ | $1683{\pm}869$ | $\mathbf{1919{\pm}794}$ | $2640{\pm}454$ |

Table 1: **Cross-physics.** Results on transferring a policy trained on domain $Y$ to domain $X$. DR: domain randomization. $^\dagger$: Task successful rate is reported.

"HalfCheetah", "FetchReach", "FingerSpin", and modify the torso mass for "Walker" and "Hopper" (see details in Appendix A.1). Different tasks are sensitive for different physical parameters, e.g.,

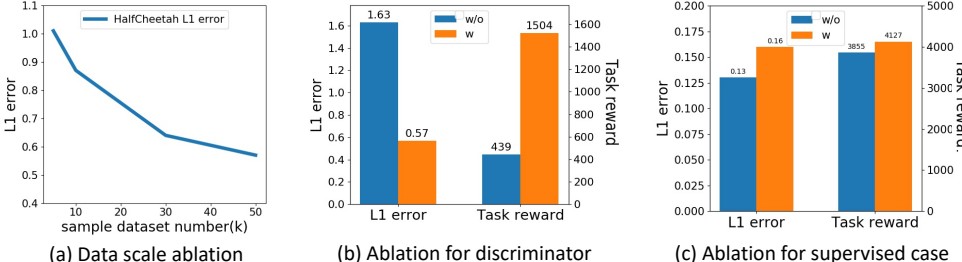

Figure 3: **Ablation study with HalfCheetah.** (a) L1 error for different dataset scale; (b) Ablation with discriminators; (c) Combining our method with supervised state estimation (using paired image-state data).

| Tasks | $L1$ Error $\downarrow$ | | | | RL Score $\uparrow$ | | | |
|---|---|---|---|---|---|---|---|---|
| | Random | Cycle-GAN | Ours | Oracle, $Y$ | Random, $X$ | Cycle-GAN | Ours, $Y{\to}X$ | Oracle, $X$ |
| HalfCheetah | 2.18 | 2.07 | **0.57** | $6270_{\pm 123}$ | $-289_{\pm 81}$ | $-119_{\pm 65}$ | $\mathbf{1504_{\pm 256}}$ | $3689_{\pm 247}$ |
| FingerSpin | 1.61 | 1.92 | **0.23** | $804_{\pm 89}$ | $0_{\pm 0}$ | $0_{\pm 0}$ | $\mathbf{341_{\pm 39}}$ | $765 \pm 68$ |
| FetchReach[†] | 0.87 | 0.94 | **0.05** | 100% | 0% | 0% | **92%** | 100% |

Table 2: **Evaluation of cross-modality alignment**, including L1 error of state estimation, and RL policy performance on the original domain $Y$ and after transferring to domain $X$. [†]: Task successful rate is reported.

while changing armature yields noticeable effect on "HalfCheetah", changing mass does not. We tackle the hard cases where physical changes matter. In this setting, we obtain the unpaired training data from two domains with a pre-trained policy in domain $Y$.

Results are shown in Table 1. The 1st column (Oracle, $Y$) reports the performance of the policy in the original domain $Y$. Directly testing this policy in environment $X$ results in significant drop in terms of RL scores (2nd column), due to the disparity in physics parameters. We also train a policy with physics domain randomization (DR, 3rd column) for direct transfer (see Appendix A.1 for details). Our method (4th column), which maps actions predicted by RL policy from domain $Y$ to $X$, demonstrates superior performance across all tasks compared to the direct deployment as well as DR baselines. We provide results of training RL in domain $X$ in the last column (Oracle $X$) as an upperbound. We also implement Cycle-GAN, which only learns random projections between actions in two domains, thus we do not report the numbers.

**Cross-modality alignment.** In this setting, we use RGB images as observations in domain $X$ and the internal state of agents as observations in domain $Y$, while keeping physics parameters the same. $G$ is then essentially a state estimator. We execute *random* actions without pre-trained policies in both domains to obtain unpaired training trajectories. As even supervised learning for state estimation with the same number of image-state pairs works poorly for "Walker" and "Hopper", we report the results on "HalfCheetah", "FetchReach", "FingerSpin" in this setting. We compute $L1$-distance between the predicted states and the ground-truth states from simulator $X$ (although we do not use them for training) as an evaluation metric. We also use RL performance as another metric: we train an RL policy in state space (domain $Y$), and test it in image space (domain $X$) by executing the predicted action based on estimated states from images.

We compare with two baselines: a random projection baseline; a image-state Cycle-GAN baseline, performing unpaired image-state translations without using dynamics (see Appendix C for details). As shown in Table 2, our approach performs significantly better than the Cycle-GAN baseline in both $L1$ error and the RL scores (the return reward), which shows the importance of incorporating the dynamics into the cycle. Note that even when the ground-truth paired (image, state) samples are provided, this is still difficult since images lie in a high-dimensional space. By exploiting the dynamics cycle-consistency, our method is able to do state estimation for transferring RL policies. We also provide results on directly training policy on the two domains (Oracle, $Y$ and Oracle, $X$).

We perform ablation studies on different elements in training with the "HalfCheetah" environment: (i) the number of training samples; (ii) the role of the discriminator. We report the results in Figure 3. It can be seen that the L1 error of state estimation reduces as training data for our dynamic cycles increases (a); training with the discriminators improves both L1 error and transferring policies by a

large margin , comparing to training without the discriminator (b), as adversarial learning with the discriminator can largely reduce the search space for finding correspondence.

We further explore our approach with supervised state estimation: Given paired image-state data, we can train a state estimator with supervised learning. We combine our dynamics cycle-consistency objective with the supervised objective to train the state estimator. As shown in Figure 3(c), We compare the results on the model using dynamics cycle-consistency ("w") and without the dynamics cycle-consistency ("w/o"). The "L1 error" measured here are on the training states which are randomly collected, the "Task reward" are from running the policy which sees states collected by the policy. We observe improvement on transferring policies with the joint model over the counterpart trained only with the supervised learning objective. In terms of L1 error of state estimation, although using paired data (strong supervision) per se will yield the best model on L1 metric (especially the state estimator in this way is trained via L1 loss. However, overfitting to the L1 loss does not completely transfer to better policy performance on downstream tasks, since the policy can observe novel states that are not in the training set. In our case, adding dynamics objective biases the L1 error by a small margin as the model cannot solely optimize for state estimation error, but it must learn dynamics as well and in turn increases the task performance. Consequently, incorporating the dynamics cycle-consistency can provide extra regularization and improve generalization on test data.

**Cross-modality-and-physics alignment.** Evaluation on two domains with different physics parameters and different input modalities. Following the cross-modality setting,

| Tasks | Oracle, $Y$ | Random | Cycle-GAN | Ours (only M) | Ours (Full) |
|---|---|---|---|---|---|
| HalfCheetah | $6270_{\pm 123}$ | $-248_{\pm 74}$ | $-226_{\pm 84}$ | $856_{\pm 385}$ | $\mathbf{1251 \pm 297}$ |
| FingerSpin | $804_{\pm 89}$ | $0_{\pm 0}$ | $0_{\pm 0}$ | $243_{\pm 54}$ | $\mathbf{305 \pm 43}$ |
| FetchReach[†] | $100\%$ | $0\%$ | $0\%$ | $92\%$ | $\mathbf{92\%}$ |

Table 3: **Cross-modality-and-physics**. Results on transferring RL policies.

we sample our unpaired training data *randomly*. We report the results of transferring RL policy in Table 3. While this setting is very challenging and the cross-modality Cycle-GAN method fails (3rd column), our method can discover the correspondence from randomly collected unpaired trajectories (last column). We perform ablation by only training to align the observations, without the translator between the actions (4th column). This shows the importance of our joint optimization approach.

**Cross-morphology alignment.** Evaluation on two domains with different morphology. We experiment with two tasks (see Appendix A.3): (i) domain $Y$ with 2-leg HalfCheetah and

| Tasks | Oracle, $Y$ | Random | Cycle-GAN | INIT | Ours, $Y \rightarrow X$ |
|---|---|---|---|---|---|
| Cheetah | $6270_{\pm 123}$ | $-250_{\pm 59}$ | $-43_{\pm 52}$ | $-37_{\pm 60}$ | $\mathbf{2471 \pm 382}$ |
| Swimmer | $366_{\pm 26}$ | $-1_{\pm 4}$ | $14_{\pm 5}$ | $-15_{\pm 3}$ | $\mathbf{204 \pm 56}$ |

Table 4: **Cross-morphology**. Results on transferring RL policies.

domain $X$ with 3-leg HalfCheetah; (ii) domain $Y$ with 3-limb Swimmer and domain $X$ with 4-limb Swimmer. In this setting, the unpaired trajectories are also *randomly* sampled and our model learns to align the observations and actions at the same time. Once the correspondence is found, we can transfer the RL policies from domain $Y$ to $X$. We compare to two baselines: one is the Cycle-GAN to perform both state-state and action-action translations, the other is using our action repetition initialization strategy before training (INIT, this baseline repeats the actions of the shared joints while uses actions for novel joints from its nearest joint, see Appendix A.4). As shown in Table 4, our approach can still perform reasonably well without finetuning the policy while the baselines completely fail. The reason behind this is that both INIT and Cycle-GAN baseline share the critical flaw: they do not take dynamics information into account. Given unpaired agent states, Cycle-GAN cannot find the correspondence between the two domains. In contrast, our method learns dynamics via dynamics cycle-consistency, which outperforms those which don't by a large margin.

## 5 REAL ROBOT EXPERIMENTS

We use an xArm robot for the *cross-modality alignment* task. The goal is to estimate the simulation states (domain $Y$) given the real robot images (domain $X$), without any paired image-state data. We *do not* have access to the internal states of the real robot. We use an uncalibrated RGB camera to capture the videos of the robot movements. We collect the real robot videos by randomly executing end-effector positional control. We collect random trajectories in xArm simulator. The training set includes 11k triplets (image, action, next image) of the real robot. We collect two testing sets from

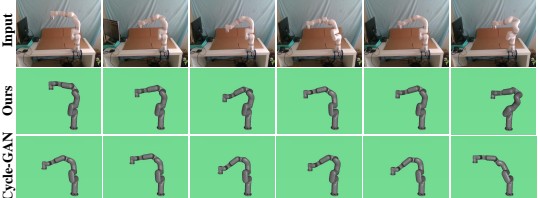

Figure 4: **Visualization** of learnt correspondence from RGB images to robot joint states with xArm robot. We render the predicted states in simulation with green background. While Cycle-GAN struggles to find the correct correspondence, the results of our method highlights the importance of dynamics cycle-consistency objective. (Best viewed in Adobe Acrobat to see the GIF of the last column.)

| Method | Random | Smooth |
|---|---|---|
| Random $G$ | 0.30 | 0.18 |
| Cycle-GAN (E) | 0.18 | 0.21 |
| Ours (E) | **0.025** | **0.033** |
| Ours (J) | 0.031 | 0.044 |

Table 5: **Real robot results.** We measure the L1 error (smaller better) of end effector position estimation. We experiment with either (i) end effector position (E), or (ii) joint positions (J) as the observations in simulator.

the real robot: a) 1,000 samples of random movement (Table 5, 1st col.), and b) 100 samples of smooth movement (Table 5, 2nd col.).

We conduct experiments using either end-effector position or joint poses (7 joint positions) as observations in simulation. Note the action is defined by the *delta movement* of the end-effector, not the exact position of the end-effector. *Thus there is no shortcuts for directly estimating the end-effector position and even harder for joint positions*. We measure the L1-distance between the predicted and ground-truth end-effector position for evaluation. We compared our method with Cycle-GAN baseline (Appendix C). As shown in Table 5, the results from Cycle-GAN is close to random and our method with dyanmics cycle-consistency achieves much lower state estimation error. Besides training with end-effector as observations (Ours (E)), we also use joint poses as observations (Ours (J)) which increase the difficulty on learning the correspondence. The reason why Ours (J) is harder is that end-effector (Ours (E)) estimation does not involve the structural information of the robot, and it is of lower dimension. The joint poses contain 7 robot joint positions. Therefore both the state mapper and the dynamic model should perform better with end-effector than joint poses, yielding better overall performance. Even so, our results are still much better than Cycle-GAN with end-effector observations. We also visualize the translation results by rendering the states in simulation in Figure 4, and observes that our state estimation results are well aligned with the real robot video.

## 6 CONCLUSION

We propose a novel framework to find observations and actions correspondence across two domains using dynamics cycle-consistency. We show the efficacy of our method on multiple downstream applications in both simulation and on a real robot. While previous approaches relies on paired data or RL polices on collecting the data for learning, we provide a *general* framework that can learn correspondence from randomly sampled, unpaired data, independent of the defined RL task. This allows the correspondence to be generalized to diverse downstream applications.

**Acknowledgements.** We like to thank Jeannette Bohg for helpful discussions on this project. This work was supported, in part, by grants from DARPA, NSF 1730158 CI-New: Cognitive Hardware and Software Ecosystem Community Infrastructure (CHASE-CI), NSF ACI-1541349 CC*DNI Pacific Research Platform, research grants from Berkeley DeepDrive and SAP, and gifts from Qualcomm and TuSimple.

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

## A EXPERIMENT SETTING

### A.1 CROSS-PHYSICS AGENT SETTING

The physical parameters in cross-physics settings are shown in Table 6. The "Parameter" column represents which type of parameter is changed in each task. Recall that we train the RL policy in domain $Y$ and evaluate in domain $X$ in our evaluation. The numbers in the columns of domain $X$ and domain $Y$ are the physics values for each domain. The physics parameter in domain $Y$ is defined by default in the OpenAI Gym MuJoCo simulation environment. The physics parameter in domain $X$ is selected so that it can cause obvious distortion to the policy trained in domain $Y$ (as shown in Table 1). Note that we did *not* perform physics parameter search for improving our method. For the domain randomization baseline, the parameter value for each episode is uniformly sampled from the range shown in the last column.

| Parameter | Envs | Domain $X$ | Domain $Y$ | DR Range |
|---|---|---|---|---|
| Armature | HalfCheetah | 0.3 | 0.1 | [0.2, 0.4] |
| | FingerSpin | 2.0 | 0.0 | [1.0, 3.0] |
| | FetchReach | 3.0 | 1.0 | [2.0, 4.0] |
| Torso Mass | Walker | 0.4 | 1.0 | [0.0, 0.8] |
| | Hopper | 1.2 | 1.0 | [0.8, 1.6] |

Table 6: **Cross_physics physical hyperparameter settings.** The parameters in domain $X$, domain $Y$ and parameter range for domain randomization baseline for each task.

### A.2 CROSS-MODALITY AGENT SETTING

In this setting, the resolution of the image observation in domain $X$ is $256 \times 256$ and we concatenate three images (current and previous two frames) as the observation input for $G$ to estimate the state. By concatenating multiple images instead of one, it allows the representation to capture the velocities and motion of the agent, which is a common practice for vision-based RL.

### A.3 CROSS-MORPHOLOGY AGENT SETTING

We introduce two tasks for the cross-morphology experiments including the "HalfCheetah" and "Swimmer" environments. As shown in Figure 5, for "HalfCheetah", we modify the agent by adding one more hind leg of the same structure as the original hind leg to obtain a three-leg cheetah. For "Swimmer", we add one more limb cloned from the original third limb, leading to a four-limb swimmer.

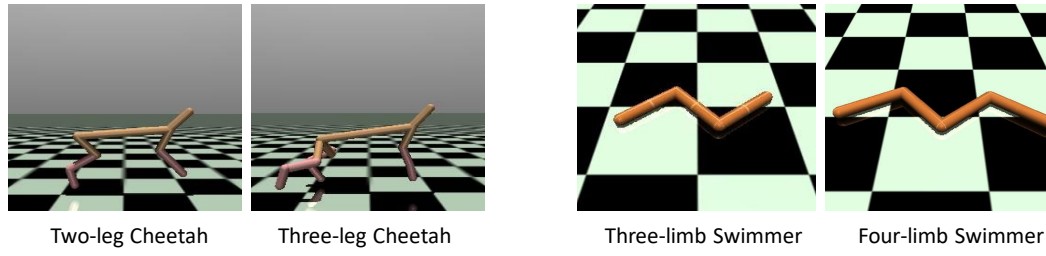

Two-leg Cheetah   Three-leg Cheetah   Three-limb Swimmer   Four-limb Swimmer

Figure 5: **Cross-morphology agent introduction.** Left: two-leg Cheetah and its three-leg counterpart. Right: three-limb swimmer its four-limb counterpart. *Please check out our supplementary video for visualization.*

### A.4 INITIALIZATION FOR ACTION ALIGNMENT MODEL

In the cross-modality-and-physics alignment, for agents with the same morphology and structure, we initialize the action translation between two domains by identity mapping, and only train the observation alignment model ($G$) in the beginning. Then we unfreeze the action alignment model ($H$ and $P$) and jointly train all models by our alternative training procedure.

For agents of different morphology (e.g., different number of limbs and joints), the dimensions of action spaces are naturally different. Thus it is impossible to initiate action mapping functions with identity mapping. Note that we can still use identity mapping for the original joints and limbs between two domains. For extra joints and limbs, we borrow the mapping function from nearby joints to initialize the novel joints. For example, in our experiments, we can find correspondence between the three-leg cheetah and the two-leg cheetah. For the newly-added leg, we use and repeat the nearby original hind leg actions to initialize its actions. There is no correspondence established from this initialization, as shown by the experiment results—we provides this initialization baseline for cross-morphology policy transfer in Table 4. This baseline perform much worse than our approach.

## B   IMPLEMENTATION DETAILS

### B.1   NETWORK ARCHITECTURE

The network architectures for each separate settings are as follows:

**Cross-physics.**   The discriminator $D$ is a five-layer MLP (hidden size: 32, 64, 128, 32) which takes states as input and predicts whether a state is true or fake. The forward dynamics model $F$ is a four-layer MLP (hidden size: 64, 128, 32) which takes current state and action as input and predicts the next state. The observation alignment function $G$ is an identity mapping. The action alignment functions $H$ and $P$ are MLPs (hidden size: 32, 64, 128, 32) which take current state and action as input and predicts corresponding action in the other domain.

**Cross-modality.**   The discriminator $D$ and the forward dynamics model $F$ are the same as that in the "cross-physics" setting. The observation alignment function $G$ is a ResNet-18 and followed by an MLP head (hidden size: 256, 64, 32) which outputs corresponding state. The action alignment functions $H$ and $P$ are identity mapping.

**Cross-morphology.**   Newtorks $D, F, H$ and $P$ are the same as that in the "cross-physics" setting. The observation alignment function $G$ is the same as that in the "cross-modality" setting.

### B.2   TRAINING

Given the dataset of unpaired and unaligned samples from two domains, we first train the forward dynamics model $F$ on domain $Y$ until it converges, we use Adam optimizer (Kingma & Ba, 2014) with initial learning rate 0.001 which is decreased by half every three epochs and train the network for 20 epochs. This is the same for all the settings. Secondly, we train the action alignment functions $H, P$ or observation alignment function $G$ by optimizing Eq. 5. The pipeline and optimization chain for each separate settings are as follows:

**Cross-physics.**   We optimize the alignment functions $H, P$ here. We set $\lambda_0 = 200$, $\lambda_1 = 1$, and $\lambda_2 = 0$ in Eq. 5, where $\lambda_2 = 0$ means we are not using the observation alignment function $G$. Note that although the forward dynamics model $F$ is involved in the back-propagation, we are fixing the weights of $F$. We train the models by using the Adam optimizer for 50 epochs with a batch size of 32. The learning rate is set to $0.001$ and decreased by $1/3$ for every 10 epochs.

**Cross-modality.**   We optimize the observation alignment function $G$. We set $\lambda_0 = 200$, $\lambda_1 = 0$, and $\lambda_2 = 3$ in Eq. 5, where $\lambda_1 = 0$ means we are not using the action alignment function $H, P$. Similarly, the forward dyanmics model $F$ is freezed. We train the model with Adam for 50 epochs with a batch size of 32. The learning rate is set to $0.001$ and decreased by $1/3$ for every 10 epochs.

**Cross-morphology (Joint training).**   Networks $H$, $P$, and $G$ are optimized by Adam optimizer while the forward dyanmics model $F$ is freezed. We combine the training procedure in cross-physics and cross-modality settings following Algorithm 1. The model is trained for 10 epochs with a batch size of 32. The training alternates every 5000 steps (e1 and e2). Note each epoch contains more training steps in joint training. The learning rate is set to $0.0001$.

### B.3   REFERENCE POLICY

We use DDPG (Lillicrap et al., 2015) with HER (Andrychowicz et al., 2017) for "FetchReach" and TD3 (Fujimoto et al., 2018) for other environments. For DDPG, we train the policy for 50 epochs of 400 episodes in each epoch. The policy exploration epsilon ratio is 0.3 and the reward discount factor

is 0.98. For TD3, we train the policy for 400k time steps. The initial exploration step is 25k. The reward discount factor is 0.99, the target network update rate is 0.005 and exploration noise standard deviation level is 0.1. When evaluating the performance of the policies, we calculate 50 episode rollout rewards across 5 different seeds. We report the rewards with variance in all tables.

## C  DETAILS OF CYCLE-GAN BASELINES

Here we introduce the details of Cycle-GAN baseline for each different settings.

**Cross-physics.**   In this setting, the Cycle-GAN baseline translates actions between two domains. The generators and discriminators are MLPs and the cycle loss is the L1 loss. The MLP network structures follow our approach. The results show that the policy fails to perform across domains and the mapping is close to random, so we have not included this baseline in the table.

**Cross-modality.**   In this setting, the Cycle-GAN baseline translates between image and its corresponding state. The image-to-state generator is the same as $G$. The state-to-image generator is composed of MLPs and six upsampling blocks. Each block consists of one transposed convolution layer, one BatchNorm layer and one ReLU activation layer. The cycle loss is L1 loss. This setting is also applied to our real robot experiment.

**Cross-morphology.**   The Cycle-GAN model translates states and actions between two domains. All generators and discriminators are MLPs and the cycle loss is L1 loss.

We visualize in Figure 6 the learned correspondence from the **cross-modality alignment** on the HalfCheetah experiment (Table 3), where observations of images are translated to states. In each plot, x-axis represents the true-state of the image and the y-axis represents the translation result from the network. Each point in the plot represents a random sample. We can see that our method is able to translate the image to the correct states as most of the dots are in the diagonal line ($y = x$), while Cycle-GAN yield near random translation similar to the random baseline. The result underlines the significance of incorporating dynamics into cycle-consistency.

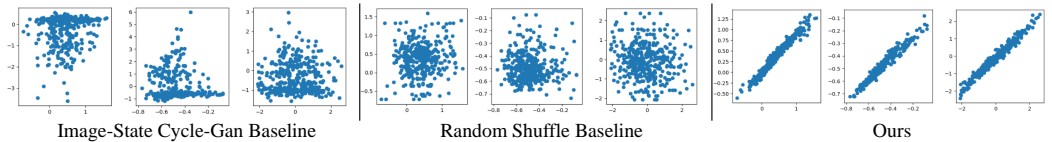

Image-State Cycle-Gan Baseline          Random Shuffle Baseline          Ours

Figure 6: **Correspondence visualization for Cycle-GAN baseline and ours.** Cycle-GAN model performs nearly the same as the random shuffle baseline while our model can correctly find the correspondence between the image modality and the state modality.

The qualitative visualizations for Cycle-GAN and ours are shown in Figure 7 for the **cross-modality alignment** on HalfCheetah. These five sub-figures from left to right in sequence are as follows: (a) One random image observation sample from the dataset. (b) The rendered image from the state which is predicted by Cycle-GAN model. (c) The rendered image from the state which is predicted by our model. (d) The image with a new renderer for our model prediction, giving a different rendering view of the state. (e) The training curve of L1 error for self-supervised state estimation. After training, our model output is almost the same as the ground-truth while the Cycle-GAN model completely fails.

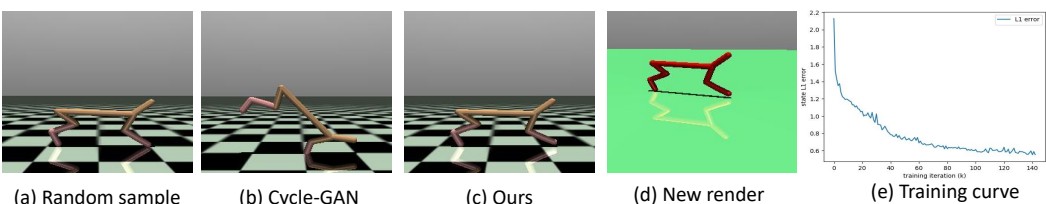

(a) Random sample          (b) Cycle-GAN          (c) Ours          (d) New render          (e) Training curve

Figure 7: **Qualitative visualization for Cycle-GAN baseline and ours.** The rendered image (c) from our model prediction looks almost the same like the original input image (a) while Cycle-GAN baseline (b) fails. The last small figure visualizes the L1 error between the ground-truth state and our model prediction during the training process and proves the effectiveness of our method in self-supervised state estimation. More visualizations are presented in the project page link given in the abstract.

