# OpenReview forum: "Learning Cross-Domain Correspondence for Control with Dynamics Cycle-Consistency"
_ICLR.cc/2021/Conference — ICLR 2021 Oral_

### Official Review · AnonReviewer4 · 2020-10-23
**Strong contribution to a hard, important problem with many applications**

**Rating:** 10
**Confidence:** 4

**Review:**

##### Summary

This work is concerned with learning mappings between pairs of domains that may differ in representation (imagery vs. configuration state), physical parameters, morphology, or combinations thereof, relying only on unpaired training data from both domains.  The key idea is to leverage the (existing) concept of cycle consistency, incorporating dynamics to formulate a cycle loss that spans vastly different domains.  The cycle is closed by learning a domain translator $G$ that maps states (the paper calls them observations - this is unimportant as the system makes no such distinction, as either domain might represent a "state" or an "observation") from domain $x$ to domain $y$, and a forward model $F$ that operates in the $y$ domain.  In addition, functions $H$ and $P$ are learned that map actions $a$ and $u$ between their respective domains $x$ and $y$, in either direction.  The difference between $G(x_{t+1})$ and $F(G(x_t), u_t)$ is then the part of the overall loss function that closes the loop. $G$, $H$ and $P$ are trained adversarially using cycle consistency losses; $F$ can be trained by regression, exploiting training sequences in the $y$ domain.

The system is evaluated on four different tasks from OpenAI Gym, and is shown to be effective in diverse settings.

##### Strengths

* This work addresses an important, fundamental problem that arises in many applications, including state estimation and sim2real transfer.
* The paper makes a significant, nontrivial contribution to this problem, allowing hard cross-domain transfer problems to be solved dramatically better than before.
* The work is well related to prior literature.
* The claims are supported by the empirical results.
* The paper is well organized, readable and clear.
* The appendix provides a lot of additional detail important for reimplementing this method and replicating the results.

##### Weaknesses

* The implementation details, in particular the neural-network architectures, appear to be very specifically tailored to the scenarios of evaluation. Instead it would be helpful to provide insight into how a user of this method should go about designing the networks, given their application domains.
* For cross-modality alignment, paired training data are typically available.  How can the method take advantage of this? How would this impact the results, e.g. those of Table 2?  Some related results are given in Fig. 3(c) but it is unclear what exactly was done and what exactly the graph shows.
* An obvious and explicitly-stated application of this work is sim2real transfer, but no such experimental results were provided (I realize that it is not easy to set up such an experiment from which strong conclusions can be drawn).

##### Recommendation

Unless I am missing important similar work, the paper makes a strong contribution in an important area.  It should be accepted.  While I wrote more text under Weaknesses than under Strengths, the weaknesses are minor compared to the strengths.

##### Questions for the Rebuttal

* The results on cross-physics alignment (Table 1) seem to leave a lot of room for further improvement.  What is the limiting factor?
* Please comment on the listed weaknesses.

##### Details

* "RL score" = return?
* There are many little typos.

---

> ### Author Response · Authors · 2020-11-20
> **Response to review #4**
>
> We thank the reviewer for the positive assessment and helpful feedback. We will address each of your comments in the following.
>
> ***Q1: “The implementation details, in particular the neural-network architectures, appear to be very specifically tailored to the scenarios of evaluation.”***
>
> **A1:** We would like to point out that our network architectures are very much simple and standard. The network is a ResNet-18 to process image input; we use the same architecture of 4-layer MLP to process vector input across all the models and applications. Consequently, there is no specifically tailored network for each scenario.
>
> ***Q2: “For cross-modality alignment, paired training data are typically available. How can the method take advantage of this? How would this impact the results, e.g. those of Table 2? Some related results are given in Fig. 3(c)...”***
>
> **A2:** Our experiments on Fig.3(c) discusses the case where a small amount of paired data is available. The notations “w/” and “w/o” in Figure 3(c) means “w/ dynamics cycle-consistency” and “w/o dynamics cycle-consistency”. The “L1 error” measured here are on the training states which are randomly collected, the “Task reward” are from running the policy which sees states collected by the policy. Thus the distributions of two sets of states are not completely aligned.
>
> In terms of L1 error of state estimation, we agree that paired data (strong supervision) per se will yield the best model on L1 metric (especially the state estimator in this way is trained via L1 loss. However, overfitting to the L1 loss does not completely transfer to better policy performance on downstream tasks, since the policy can observe novel states that are not in the training set. In our case, adding dynamics objective biases the L1 error by a small margin as the model cannot solely optimize for state estimation error, but it must learn dynamics as well and in turn increases the task performance.
>
> We address a similar question from R1 with this answer here. We have highlighted the explanation in red in the revised paper.
>
> ***Q3: “An obvious and explicitly-stated application of this work is sim2real transfer, but no such experimental results were provided (I realize that it is not easy to set up such an experiment from which strong conclusions can be drawn).”***
>
> **A3:** We thank the reviewer for the great suggestion. In fact, our real robot experiment can be viewed as the first step of the sim2real transfer experiment. Once we have the current cross-modality correspondence between the real image and simulation states, we can potentially train a RL policy in the simulation and apply it directly to the real robot. However, due to the limitation of time and access constraints to the robot, we cannot provide the experiment results in the rebuttal period. We believe our method can be extended to the sim2real applications in the future since our alignment model is accurate and  lightweight which allows it to work in real time.
>
> ***Q4: “The results on cross-physics alignment (Table 1) seem to leave a lot of room for further improvement. What is the limiting factor?”***
>
> **A4:** Indeed it is a very challenging problem and there is still a lot of room to improve. We have considered two potential limitations and directions for our approach:
>
> (1) The current forward dynamics model F only considers 1 step prediction, we can extend the model to obtain more accurate dynamics by training for predicting multiple future steps. Similarly, our dynamics cycle could be possibly extended to multiple time steps.
>
> (2) Another potential direction is to incorporate and explore more structural information related to the environment during learning. For example, we can adopt graph neural networks or attention-based models to model agent morphologies and the agent-environment interactions for better generalization.
>
> ***Q5: “Details: "RL score" = return? There are many little typos.”***
>
> **A5:** Yes, “RL Score” = return, we have revised it along in the paper. We will continue refining the writing and fixing the typos.

---

### Official Review · AnonReviewer2 · 2020-10-26
**A somewhat small yet solid contribution; paper clarity could improve**

**Rating:** 7
**Confidence:** 3

**Review:**

This paper presents a technique that leverages cycle-consistency to align data across domains given only samples of one-step trajectories (x_t, a_t, x_{t+1}) from each. The cycle-consistency is used over the space of actions across two domains: e.g., given data from X [in the form (x_t, a_t, x_{t+1})] and data from Y [in the form (y_t, u_t, y_{t+1})], the system should be able to discover an "alignment" between states across the two domains by enforcing that the action spaces be mappable to one another (either identical, as in the "cross-modality only" applications, or via cycle-consistency, as in the "joint model" applications). The authors go on to show that their approach, among other things, allows for strong alignment between real-world images and their corresponding underlying state.

Overall, the paper presents some solid results across a handful of interesting domains/applications. The paper is occasionally difficult to follow, particularly on the first read-through, in part because it is unclear precisely what form the data takes until Section 3, a problem not alleviated by a somewhat-unclear Figure 1 (see more detailed comments on this below). The introductory paragraph beginning with "In this paper" is quite hard to follow. Providing some more concrete examples in the introduction would be incredibly helpful for clarity. For example, mentioning that the input data to the learning algorithm takes the form of 3-element tuples (and their components), would be helpful.

The authors might also consider combining and aligning Figures 1 & 2 to help the reader follow along with the early pages of the paper. Perhaps also specifying the domain (i.e. functional domain: the space of the variables) on the figures themselves would help here. This lack of clarity in the beginning of the paper is likely its biggest weakness, as it is otherwise a good theoretical idea supported by strong results.

Smaller comments:
- Though it is clear why training _all_ of the networks jointly would lead to mode collapse (motivating the need to train F separately), it is not made particularly clear why the remainder of the system needs to be trained using the 'alternating' procedure employed throughout. A sentence or two explaining this decision would be helpful.
- Particularly in the figures, the authors should make clear that the `y` images are in fact renders corresponding to the predicted state. Without this "disclaimer" readers coming from the purely-image-driven cycle-consistency community may mistakingly believe that those images are the direct output of the CycleGAN baselines or the proposed approach. Modifying the green-backgound images in Fig. 1(a) to include "Render of y_t" (or similar) would help avoid such a confusion.
- Figure 1(b) is potentially misleading, since really the images are assumed to be identical between the two domains. Additionally the arrows on (b) and (c) in fig. 1 are also somewhat misleading, since it is not clear what the arrows represent. Consider adding more annotations on this figure describing in more depth what this figure is trying to communicate: namely the types of applications enabled by the approach, rather than the structure of the proposed approach itself.
- The Related Work section is quite thorough and helpful for providing context. Two other papers that the authors might have missed and consider citing that made progress in solving similar tasks are (1) "Adapting Deep Visuomotor Representations with Weak Pairwise Constraints" (Tzeng, Devin, et al, 2016) [which uses a task-specific loss for aligning sim/real robot arm images] and (2) "GeneSIS-RT: Generating Synthetic Images for training Secondary Real-world Tasks" (Stein & Roy, 2018) [which uses cycle-consistency for task-agnostic sim-to-real and then uses the translated data for quadcopter flight]. Both papers circumvent the need to collect paired sim and real images for real-world robotics applications.

---

> ### Author Response · Authors · 2020-11-20
> **Response to review #2**
>
> We thank the reviewer for the positive assessment and helpful feedback. We thank the reviewer’s comment on writing especially about the introduction and Figure 1. We have made some initial edits to the introduction and Figure 1 in the revised version of the paper. We will make more edits after the rebuttal period to avoid confusion.
>
> ***Q1: “Though it is clear why training all of the networks jointly would lead to mode collapse (motivating the need to train F separately), it is not made particularly clear why the remainder of the system needs to be trained using the 'alternating' procedure.”:***
>
> **A1:** We find the alternating procedure stabilizes the training. Note that the forward dynamics model F is still a neural network rather than the ground-truth environment, so there might be states that are not yet explored by F. Therefore if both action and state mappers are free to change, they can potentially act together to fool F by exploiting its out-of-distribution regions. In this way, the learning objective can still be minimized but the output of state and action mappers does not form a valid dynamics transition. We find by alternating the training of both networks instead of training all in one time can alleviate this potential issue.
>
> ***Q2: “Related Work”***
>
> **A2:** Thanks for the suggestions on the reference paper, we have added them with discussions in our related work in the revised paper.

---

### Official Review · AnonReviewer1 · 2020-10-28
**Novel and interesting work**

**Rating:** 8
**Confidence:** 3

**Review:**

Summary:
The paper provides an interesting (and to my knowledge, novel) approach for learning a mapping of actions and observations from one domain/character to another reasonably similar domain/character. This mainly allows the transfer of skills (i.e. policies) from one domain/character to another. Immediate use cases of this approach are in imitation learning and sim2real transfer. In order to learn these mappings, the authors use the idea of cycle-consistency in generative adversarial networks and adapt it to the imitation learning task. Importantly, this allows them to obviate the need for paired state samples across domains. The authors show that their method not only works across modalities (vision from real robot to states in simulated robot), but it also works to some extent on different character morphologies.

Reasons for score:
Overall, I vote for accepting this paper. To my knowledge, the method is novel and provides a viable and interesting approach for imitation learning and sim2real transfer. The experiments and the final quality are also high, however, the broader applicability to more challenging tasks and its limitations remain to be seen.

Cons:
- The chosen tasks are (understandably) simple, therefore the applicability of the methods to more challenging environments remain to be seen.
- The limitations of the method are not discussed well. I believe some commentary on the challenges of this method (e.g. the ease of training with GANs) would be useful. Also, more discussion on the use cases of the method and where it excels existing methods is missing.
- The code is not provided which hampers reproducibility. I strongly suggest providing the code if possible.

Question:
- How does the method handle partial observability? In the robot arm example, a single image of the robot does not contain information about the velocity and angular velocity (of the joints). I'm confused as to how the model can actually infer these or work without knowing them.
- Can you spend more time explaining Figure 3 (c)? It seems strange that the "L1 error" increased when more data was available.
- Related to the last question: in this case, can you pre-train G the same as F and use cycle-consistency only for H and P? A brief explanation would suffice.


Fixes or suggestions:
- I will have to double check the conference style guides, but having tables and algorithms intertwined with the text in a single-column publication is not pleasing to the eyes. I suggest rearranging these elements.
- The plots in Figure 3 can be improved. At the very least, the distortions caused by resizing the images should be addressed.

---

> ### Author Response · Authors · 2020-11-20
> **Response to review #1**
>
> We thank the reviewer for the positive assessment and helpful feedback. We will address each of your comments in the following.
>
> ***Q1: “The chosen tasks are (understandably) simple.”***
>
> **A1:** We believe that our work can be expanded to a large range of tasks including motor control tasks and robot arm manipulation tasks across multiple transfer settings. The tasks we presented in the paper are typical ones from these different scenarios respectively.
>
> ***Q2: “The limitations of the method are not discussed well. [...] some commentary on the challenges of this method [...]”***
>
> **A2:** Indeed there are still a lot of spaces to improve for our approach. We discuss two potential future directions for extending our method:
>
> (1) Current version of our model can only afford translation between two domains, i.e., the model needs to be retrained for a new domain. One future work can be designing a unified model that can automatically choose and adjust modes when a new domain is present.
>
> (2) Current version learns dynamics cycle-consistency at a single next time step. Future work can expand the model to multiple time steps to learn long horizon dynamics.
> We have highlighted the discussion in the conclusion section in our revised paper.
>
> ***Q3: “Also, more discussion on the use cases of the method and where it excels existing methods is missing.”***
>
> **A3:** Please see our discussion on the significance of our model compared with previous ones in the related work section. In summary, previous works usually presume knowledge of downstream tasks or the presence of a pretrained RL policy, hence its use case is limited. In contrast, our method can learn the correspondence between simulated and real robots through unpaired and randomly collected trajectories. While previous works that learn cycle-consistency are generally limited to visual input, ours extends to dynamics and structural domains.
>
> ***Q4: “The code is not provided which hampers reproducibility.”***
>
> **A4:** We agree with the reviewer that it is important to provide the code for reproducibility. We are currently working on the release of our code.
>
> ***Q5: “How does the method handle partial observability? In the robot arm example, a single image of the robot does not contain information about the velocity.”***
>
> **A5:** We concatenate frames within a short time horizon (current frame plus two previous frames in our experiments) as input so that velocity can be inferred by the model. It is a common practice for vision-based reinforcement learning. We have included more detailed explanations in Appendix A2.
>
> ***Q6: “Can you spend more time explaining Figure 3 (c)? It seems strange that the "L1 error" increased when more data was available.”***
>
> **A6:** We would like to first clarify that the “w/” and “w/o” in Figure 3(c) means “w/ dynamics cycle-consistency” and “w/o dynamics cycle-consistency”. The “L1 error” measured here is on the training states which are randomly collected, the “Task reward” is from running the policy which sees states collected by the policy. Thus the distributions of two sets of states are not completely aligned.
>
> In terms of L1 error of state estimation, we agree that paired data (strong supervision) per se will yield the best model on the L1 metric (especially the state estimator in this way is trained via L1 loss. However, overfitting to the L1 loss does not completely transfer to better policy performance on downstream tasks, since the policy can observe novel states that are not in the training set. In our case, adding dynamics objective biases the L1 error by a small margin as the model cannot solely optimize for state estimation error, but it must learn dynamics as well and in turn increases the task performance.
>
> We address a similar question from R4 with this answer here. We have highlighted the explanation in red in the revised paper.
>
> ***Q7: “Related to the last question: in this case, can you pre-train G the same as F and use cycle-consistency only for H and P?”***
>
> **A7:** With the above discussion, G can be initialized by pre-training using the supervised paired data, but we believe that it cannot be frozen thereafter like F, as jointly training G in the context of dynamics cycle-consistency benefits the downstream tasks that involve dynamics, which is our ultimate goal. As for the setting in Figure 3 (c), which is the cross-modality alignment setting, the dynamics are the same across the domain so there are no needs for training H and P. Finally, we want to emphasize again that in all our general settings, we assume there are no paired state or image-state data available.
>
> ***Q8: “Fixes or suggestions”***
>
> **A8:** We thank the reviewer for the suggestions. We have fixed the resizing problem in Figure 3. We will rearrange the tables for the camera-ready version if accepted. Currently, we are maintaining the original format for the convenience of reviewing.

---

### Official Review · AnonReviewer3 · 2020-10-31
**Review for "Learning Cross-Domain Correspondence for Control with Dynamics Cycle-Consistency"**

**Rating:** 6
**Confidence:** 2

**Review:**

** Paper Summary **

This paper proposed a novel framework to establish the correspondences across observations and actions, by using dynamic cycle-consistency. The paper focused on robotics applications, but the technical solutions are based on a pioneering method computer vision, generative adversarial networks (GANs) and cycle-consistency constraints, whose robustness has shown in many previous literatures. The proposed method was applied and attained the robustness on various tasks, such as cross-physics alignment, cross-modality alignment, cross-modality-andphysics alignment.

** Paper Strength **
+ The paper is well organized and written.
+ Borrowing the concepts of GANs and cycle-consistency to solve the unpaired problems in cross-modal correspondence makes sense and good performance gains are expected.
+ Solving several simulation-based settings and real robot settings was nice.

** Paper Weakness **

I have no major comments on this paper, but have some minor ones.
- In Table 5, Ours (E) and Ours (J) have shown the performance gaps. It would be great if the reasons are properly mentioned.
- In Table 4, the authors compared the quantitative results with Cycle-GAN and INIT. It would be great if the thorough discussion for these experimental settings are provided.
- Computational complexity analysis is also required.

---

> ### Author Response · Authors · 2020-11-20
> **Response to review #3**
>
> We thank the reviewer for the positive assessment and helpful feedback. We will address each of your comments in the following.
>
> ***Q1: “In Table 5, Ours (E) and Ours (J) have shown the performance gaps. It would be great if the reasons are properly mentioned.”***
>
> **A1:** Estimating the end-effector (Ours (E)) is an easier task compared to estimating the joint poses (Ours (J)). The reason is that end-effector (Ours (E)) estimation does not involve the structural information of the robot, and it is of lower dimension. The joint poses contain 7 robot joint positions. Therefore both the state mapper and the dynamic model should perform better with end-effector than joint poses, yielding better overall performance. We have revised our paper and highlighted the explanation in red.
>
> ***Q2: “In Table 4, the authors compared the quantitative results with Cycle-GAN and INIT. It would be great if the thorough discussion for these experimental settings are provided.”***
>
> **A2:**  In Table 4 we list cross-morphology experiments, i.e., agents of different structures. INIT baseline repeats the actions of the shared joints while uses actions for novel joints from its nearest joint (details in Appendix A4). Both INIT and Cycle-GAN baseline share the critical flaw: they do not take dynamics information into account. Given unpaired agent states, Cycle-GAN cannot find the correspondence between the two domains. In contrast, our method learns dynamics via dynamics cycle-consistency, which outperforms those which don’t by a large margin. We have highlighted the explanation in red in the revised paper.
>
> ***Q3: “Computational complexity analysis is also required.”***
>
> **A3:** To have an estimation of the computational complexity, please check the implementation details of our method in Page 6 and Appendix B.1. Our models (observation, action alignment models and forward dynamics model) have ResNet-18 to process image input, and 4-layer MLPs to process vector input. They are light-weighted and can be computed extremely fast (in ms) on GPUs. Our method can be deployed in real-time.

---

### Public Comment · ~Nam_Hee_Gordon_Kim1 · 2020-11-12
**Previous Work on Dynamical System Correspondence**

Dear Authors,

Congratulations on the excellent work.

I would like to point you to a previous work on cross-system motion correspondence that uses a very similar framework. It uses learned latent forwrad dynamics model as well as autoencoders trained with cyclic consistency loss.

http://proceedings.mlr.press/v120/kim20a.html
Kim, N.H., Xie, Z. & van de Panne, M.. (2020). Learning to Correspond Dynamical Systems. Proceedings of the 2nd Conference on Learning for Dynamics and Control, in PMLR 120:105-117

While your specific subproblem within motion correspondence is different, your work seems to offer improvements upon the limitations of our work, namely supporting action correspondence as well as using random trajectories instead of expert trajectories.

---

### Decision · Program_Chairs · 2021-01-07
**Final Decision**

**Decision:**

Accept (Oral)

**Comment:**

The paper proposes a new solution for cross-domain correspondence in control, which combines GANs and cycle-consistency, and separates shifts in observation space and in action space. The paper targets unpaired data / simulations, and discovers alignment of state by enforcing that domains are mappable.

The paper was received well by reviewers, who pointed out several strengths: a strong contribution on a fundamental problem, and an interesting formulation; a well written and well positioned paper; This compensates minor weaknesses, in particular the fact that transfer has been tested between two different simulated environments.

The reviewers unanimously suggested acceptance, the AC concurs.